# Electromyographic Response of the Abdominal Muscles and Stabilizers of the Trunk to Reflex Locomotion Therapy (RLT). A Preliminary Study

**DOI:** 10.3390/jcm11133866

**Published:** 2022-07-03

**Authors:** Fátima Pérez-Robledo, Juan Luis Sánchez-González, Beatriz María Bermejo-Gil, Rocío Llamas-Ramos, Inés Llamas-Ramos, Antonio de la Fuente, Ana María Martín-Nogueras

**Affiliations:** 1Department of Nursing and Physiotherapy, University of Salamanca, 37008 Salamanca, Spain; fatima_pr@usal.es (F.P.-R.); beatriz.bermejo@usal.es (B.M.B.-G.); rociollamas@usal.es (R.L.-R.); inesllamas@usal.es (I.L.-R.); anamar@usal.es (A.M.M.-N.); 2Department of Physiology and Pharmacology, Institute of Neurosciences of Castilla and León (INCyL), University of Salamanca, Avenida Alfonso X El Sabio s/n, 37007 Salamanca, Spain; jfuente@usal.es

**Keywords:** electromyography, muscle activity, neurorehabilitation, Vojta therapy, tactile stimulation

## Abstract

Reflex locomotion therapy (RLT) was developed by Vaclav Vojta in 1954 as a diagnostic and treatment tool. This therapy is mainly used to rehabilitate children with motor disorders and risk of cerebral palsy. It is also used for adults with neurological and motor impairment. RLT is based on specific postures and regular stimulation points through which a series of reflex responses are triggered. The neurophysiological mechanisms of this therapy have recently been discovered. This study aims to objectively evaluate muscular responses at the abdominal level after stimulation in the first phase of reflex rolling by showing, with surface electromyography analysis (sEMG), the muscular activity in trunk stabilizing muscles (rectus abdominis, external oblique, internal oblique, and serratus anterior) before, during, and after the application of RLT. A total sample of 27 healthy subjects over 18 years of age was recruited. An experimental study on a cohort was conducted. Two experimental conditions were considered: stimuli according to the Vojta protocol, and a control non-STI condition. Regarding muscular electrical activity, statistically significant differences were determined in all muscles during right-sided stimulation in the VSTI condition (*p* < 0.001), but not in the non-STI condition. The mean increase in muscle activity in the VSTI condition during the first stimulation ranged from 7% to 20% in the different abdominal muscles. In conclusion, an sEMG response was observed in the abdominal muscles during stimulation of the pectoral area as described in RLT, compared to stimulation of non-described areas.

## 1. Introduction

Reflex locomotion therapy (RLT) was developed by Vaclav Vojta in 1954 as a diagnostic and treatment tool. The innate behavior patterns of each animal species have been studied since the 1960s. Different studies [1,2] have shown that an essential part of our behavior is preprogrammed. In other words, there is innate coordination. The novelty of Reflex locomotion therapy (RLT) refers to a genetic program specific to the human species, describing the “human motor ontogenesis” [3].

Reflex locomotion, or Vojta therapy, includes two major coordination complexes: reflex crawling and reflex rolling. Both complexes contain three inseparable components: postural activity, righting mechanisms, and phasic motor skills [4].

RLT is based on specific postures and regular stimulation points through which a series of reflex responses are triggered. Dr. Vojta first described reflex crawling, and later reflex colling, with its two different phases [5].

The reflex rolling complex is represented in the ontogeny development. It consists of four phases, from a supine position to a lateral position, ending in a quadruped position [6].

The neurophysiological mechanisms of this therapy have recently been discovered. In their study, Sanz et al. [7] concluded that through the first phase of Vojta reflex rolling, activation of the basal ganglia, the putamen, the anterior cerebellum, and the thalamus is achieved. These structures play an essential role in motor acts. In a recent study [8], the stimulation of the pectoral area in the reflex rolling coordination complex was used, showing activation of cortical areas such as supplementary motor areas (SMA) and premotor areas (PMA) (Brodmann areas BA6 and BA8), which are areas responsible for movement planning, regulation, and execution as demonstrated by Martinek et al. [9]

This therapy is mainly used to rehabilitate children with motor disorders and infants at risk of cerebral palsy [10,11,12,13], increasing cortisol levels in this type of patient [14]. Likewise, it is also used for adults with neurological and motor impairment, especially for adults who have suffered a stroke [15]. Its applicability extends to other neurological pathologies such as multiple sclerosis [16], as RLT improves postural control and balance [17], or even to peripheral injuries, spina bifida, congenital malformations, or orthopedic problems [18,19].

Sometimes it is not possible to see the expected reflex responses during therapeutic practice; however, even if not visible, an isometric contraction of the musculature occurs. Until now, only one study has demonstrated contraction of the musculature during reflex crawling in reflex locomotion therapy, concluding that there is an activation of the deltoid muscle and rectus femoris muscle [20]. However, no study has evaluated the activity of the abdominal muscles during stimulation in the first phase of reflex rolling.

For this reason, this study aims to objectively evaluate muscular responses at the abdominal level after stimulation in the first phase of reflex rolling according to the Dr. Vojta method by showing the muscular activity in trunk stabilizing muscles (rectus abdominis, external oblique, internal oblique, and serratus anterior) before, during, and after application of Vojta therapy.

## 2. Materials and Methods

### 2.1. Design

An experimental study was conducted on a randomly selected cohort of healthy subjects aged over 18 of both sexes.

All participants were volunteers who expressed their desire to participate, with an approximate sample of 30 subjects.

Due to the lack of previous studies to obtain the necessary calculation data, an initial sample size calculation was not established in this study. It was decided to perform a post hoc statistical power calculation, assuming that a statistical power more significant than 80% is valid for acceptance of the results.

Two experimental conditions were considered: stimuli according to the Vojta protocol: a Vojta-specific tactile input-group (VSTI condition); and the control: a non-specific tactile input-group (non-STI condition). The subjects were informed about the study objectives and methodology, but were unaware of the meaning of the experimental conditions.

This trial was registered in ClinicalTrials.gov with the register number NCT04818879 and was approved by the corresponding bioethics committee (registration number: 609). Declaration of Helsinki ethical standards were followed.

### 2.2. Participants

A sample of healthy subjects over 18 years old participated with full cognitive skills who were unaware of Vojta therapy or their response to stimuli after applying the therapy. They were recruited by email and through usual channels of information in a university community.

Subjects with neuromuscular pathologies that affect the abdominal muscles, subjects with previous surgeries in the area, subjects suffering from sensory disturbances or showing presence of inflammatory disease, fever, or pregnancy, subjects who were drug users or undergoing a pharmacological treatment that could affect the nervous system, and subjects with chronic neurological or organic disorders that could alter the results were excluded.

All subjects provided informed consent.

### 2.3. Tactile Stimuli Location

In the VSTI condition, each subject was stimulated in the pectoral area during the locomotion complex of reflex rolling in its first phase. The subjects were placed in a supine position aligned on the axial axis, with the arms along the body, the lower extremities in extension, and the head extended with a rotation of approximately 30° towards the side of the stimulation. A tactile pressure between 1.4 and 1.8 kg/cm^2^ ± 200 g was performed with a dorsal, cranial, and medial directional specific tactile stimulus, toward the contralateral shoulder according other studies [7,8], for stimulation in the intercostal space between the sixth and seventh or seventh and eight ribs, depending on the participant’s chest position [18]. To locate the space, the space in which the intersection occurs of a line perpendicular to the xiphoid process and a line perpendicular to the middle of the clavicle occurs was taken into account. Figure 1 shows the stimulation point and how to locate it.

In the non-STI condition, the subjects were stimulated in areas not described by the Vojta methodology (distal third of the quadriceps and 8 cm cranial to the superior angle of the patellar bone). This area has already been used in previous studies as a “control stimulus” due to its low density of mechanoreceptors [7,8,21] and does not cause a proprioceptive stimulus.

### 2.4. Electromyographic Recording (rEMG) Stimuli

The two experimental conditions were carried out in the same 40 min session, in a properly lit and heated room. All subjects were stimulated by an expert Vojta therapist, first in the non-STI condition and then in the VSTI condition. In all subjects, the following procedure was developed:10 min of absolute rest without EMG logging;1 min of pre-stimulation rest with EMG registration (R1) non-STI condition;2 min of non-STI right side styling condition with EMG (S1) registration;1 min break with EMG (R2) non-STI condition.;2 min of left side non-STI condition stimulation with EMG (S2) logging;1 min of rest with EMG registration (R3) non-STI condition;10 min of absolute rest without EMG logging;1 min of pre-stimulation rest with EMG (R1) VSTI condition record;2 min of right-side stimulation VSTI condition with EMG recording (S1);1 min of rest with EMG registration (R2) VSTI condition;2 min of left side stimulation VSTI condition with EMG recording (S2);1 min standby with EMG (R3) VSTI condition.

### 2.5. rEMG Acquisition

The EMG equipment used was a PET4 (Brainquiry), which contains an amplifier and an analog-to-digital converter. The system was configured as a four channel plus common reference line device, and every muscle was recorded by one of the channels and the standard reference. Data from the amplifier were sent to a computer equipped with the software BioExplorer (www.cyberevolution.com) (access date 1 February 2022). This software includes a comma-separated-values exporting module to extract data for processing with external tools.

Four muscles (right external oblique (REO), left external oblique (LEO), right internal oblique (RIO), left internal oblique (LIO)) were recorded. A two centimeter diameter Ag/AgCl electrode was placed on the skin over each muscle. To measure the RIO and LIO, the electrodes were placed in a position one centimeter from the anterior superior iliac spine towards the medial; to measure the measure the REO and LEO, the electrodes were placed in the middle third of the total length of these muscles. One extra electrode of similar characteristics was placed two centimeters above and lateral to the electrode located on the REO as a reference. Electrodes were placed before the non-STI conditions and were maintained through the end of the VSTI condition. This method enables a recording changes in the electrical activity of the muscle under the electrode and has low dependence on muscle fiber direction, but it makes it difficult to compare data between muscles.

### 2.6. rEMG Data Analysis

The signals acquired were sampled at 750 Hz and filtered by an 80 Hz high-pass digital filter to reject electrocardiogram (ECG) signals, which have a high intensity at registered locations and can corrupt data. A manual review was performed at the end of the process to clean the data of spurious artifacts. Subsequently, signals were rectified; activity 5 to 15 s from the beginning was considered as the baseline and used as a reference for the recording. Data were represented as a factor of the baseline value.

For all the calculations and cases, data from 30 to 60 s were defined as the pre-stimulation phase (R1), data from 80 to 140 s were marked as the first stimulation phase (S1), data from 200 to 230 s as the second rest (R2), data from 260 to 330 s as the second phase stimulation (S2), and data from 370 to 400 s as the post-stimulation phase (R3), both in the non-STI condition and VSTI condition situations.

Response times to stimuli were also measured in the non-STI condition and the VSTI condition, marking the moment after the stimulus when the signal crossed a value corresponding to two standard deviations over the mean of the signal in the 10 s prior to the stimulus.

### 2.7. Descriptive Data Analysis

The statistical treatment was carried out with IBM-SPSS version 26. A descriptive analysis of all study variables was performed, expressed in mean (standard deviation) or median (interquartile amplitude) for the quantitative variables, and percentage and count for the qualitative ones. An intracondition inferential analysis was carried out on surface electromyography (sEMG) registration variables using non-parametric tests (Mann–Whitney-Wilcoxon test or the Wilcoxon test) once the normality of the qualitative variables was analyzed. A security level of 95% was established. Effect sizes were calculated with Hedges’ g.

The statistical power of the study was calculated from the size of the mean effect of the stimulation periods.

## 3. Results

### 3.1. Descriptive Data

The total sample consisted of 27 participants, 23 being female. The whole sample’s mean age was 20.58 ± 0.72 years. Sample features are summarized in Table 1. Sociodemographic data and level of physical activity were collected through the International Physical Activity Questionnaire (IPAQ). No adverse effects were observed in the patients during the experiment.

### 3.2. Muscular Electrical Activity

The electrical activity of each muscle was determined from the baseline of each experimental condition prior to stimulation. The changes were calculated as a factor of the baseline value. The electrical muscle activity recorded in both experimental conditions in each muscle is shown in Figure 2.

Statistically significant differences were determined in all muscles under right-sided stimulation (S1) in the VSTI condition (*p* < 0.001), but not in the non-STI condition.

Throughout the stimulation of the left side and the second stimulation, statistically significant differences were only recorded in the REO, LEO, and LIO muscles in the VSTI condition. No difference in the second stimulus was recorded over the non-STI condition.

The mean increases in muscle activity in the VSTI condition during the first stimulation (S1) were 20% in the REO muscle, 14% in the LEO muscle, 24% in the RIO, and 7% in the IOL. During the second stimulation (S2), all increases were about half of that during S1. In the non-STI condition, there were no average increases in muscle activity greater than 2% during any stimulation. Given the variability in the variables recorded, the data presented in Table 2 refer to the median so that increases corresponding to 50% of the subjects in the sample are identified.

In the VSTI condition, the differences in muscle activity between S1 and repose pre-stimulus (R1) were statistically significant in all muscles, and the differences between S2 and rest between stimuli (R2) in all muscles except IOL were also statistically significant. The muscular activity achieved in S1 was statistically higher than in S2 in all muscles. There were no statistically significant differences between response pre-stimulus and repose post-stimulus. In the non-STI condition, some statistical differences were recorded in the REO and LEO muscles between the R3 and R1 rests and the S2 and S1 stimuli (Table 3).

Effect sizes during stimulation periods were calculated, resulting in a very high effect size (d > 1) in all muscles during S1 (Table 2).

The statistical power of the study was calculated from the size of the mean effect through the S1 and S2 periods for all muscles, being 89.59%. The results of both groups are graphically explained in the Appendix A.

## 4. Discussion

The main objective of the present study was to analyze the differences between the VSTI group procedure and the non-STI group procedure in trunk stabilizing muscles’ electric muscular activity in healthy participants. Our findings highlight that specific pectoral stimulation area at the intercostal space, on the mammillary line between the seventh and eighth ribs according to Vojta therapy, activates innate muscle responses in the abdominal oblique muscles compared to a sham stimulation, as assessed by sEMG in healthy subjects. Nowadays this is the second quasi-experimental study that has recorded electromyographic activity by stimulating the pectoral area through RLT in the reflex flip position.

The obtained results shown that the groups were homogeneous in electrical activity at the beginning of the study, showing that the changes produced were due to the stimulus received. After performing the first stimulation in the VSTI, a clinically relevant increase was observed in all the muscles analyzed. In addition, the change produced was maintained during the rest period that occurred between both stimulations. Changes were observed during the second stimulation, but were smaller than in the first.

These differences were only observed in the VSTI group, so it can be said that the pectoral area generates a change in electrical muscle activity compared to the low-density point of mechanoreceptors established as a control point in the non-STI.

In addition, there were changes between the muscle activation presented at the beginning of the intervention and that presented at the end of the intervention, and although it was present in both groups, it was systematic in all the muscles in the VSTI group, unlike in the non-STI group. In this group, these changes happened in the external obliques and may even be due to the movement of the costal grill during the time in which the participant remained still, as well as due to the changes in the breathing produced by the maintenance of a posture.

Previously, Sanz et al. [8] recorded EMG activity in a clinical trial of 40 healthy subjects divided into two groups: intervention and control groups. The same stimulation points were used in our study. However, that study differed in stimulation time and the evaluated musculature; extensor wrist musculature and anterior tibial and upper rectus abdominis muscles were evaluated. The VSTI group showed greater muscle activation than the non-STI group. These differences between VSTI and non-STI with sEMG were also observed in the study of Perales and Fernández [22] in the extensor muscles of the fingers.

The present study records abdominal muscles activity to establish the effect of Vojta therapy on trunk stabilization function. This could be applied for therapeutic purposes in multiple pathologies and allows us to understand how RLT could act in postural control and ventilation, among other physiological functions.

The presence of an experimental control condition, the non-STI condition, was another of the strengths. Not all studies conducted to date have been able to make this comparison, so although they have shown a significant increase in muscle activity, the results cannot be compared with a control situation. This is the case in the studies carried out by Laufens et al. [23,24], in which they recorded the contraction of the bilateral anterior tibial and femoral biceps, femoral rectum, triceps, and biceps brachii by surface electromyography.

In another study conducted without a control group [20], upper and lower extremity muscle activation in the reflex crawling position of the RLT in healthy subjects was recorded by surface electromyography. There was activation of the femoral and deltoid muscles bilaterally during the stimulation. Therefore, it was evident that through the points stimulation described by Vojta, there is a reflex activation of the musculature, facilitating the coordination complexes such as reflex rolling and crawling.

The muscle activation observed in the present study and other studies carried out with RLT could be based on reflex responses mediated by CNS structures. Gajewska et al. [20] were the first to approximate the origin of the information during RLT and concluded that it was transmitted through the ascending and descending propriospinal tracts. Sanz et al. [7,21] have shown how stimulation of the pectoral area activates subcortical areas such as the putamen, cerebellum, or basal ganglia. Finally, Hok et al. [25] have pointed out the modulation of pontomedullary reticular formation as the structure responsible for the motor acts produced by RLT. These studies showed the neurophysiological mechanisms presented during stimulation, which are possibly responsible for the observed changes.

### Limitations

The authors are aware of the limitations of the study presented. The use of a recording electrode for each muscle implies inability to quantify the precise amount of activation in each muscle. Future studies will correct this by performing the sEMG traditionally, i.e., placing two electrodes on each muscle and accurately analyzing muscle activation.

This limitation may have been the cause of the variability between records observed in the sample, which determined the use of non-parametric tests. However, the study’s statistical power shows that, although parametric tests could not be used, the results are solid.

The differences in activation shown in the VSTI were not that evident during the second stimulus, so it is necessary to expand the research to determine if the changes do not occur because of the stimulus, or if this is due to the recording method.

For this reason, in future research, it will be necessary to evaluate whether these differences are due to electrode placement, the protocol of stimulus application, or the recording of electrical activity.

In addition, we recorded only a specific time window, of eight min of stimulation and a one-minute resting period, with sEMG. Therefore, we cannot determine or estimate more prolonged muscle pattern effects.

## 5. Conclusions

A specific sensory and proprioceptive stimulation according to Vojta Therapy in the pectoral area located in the intercostal space at the mammary line between the seventh and eighth rib activates innate muscle responses in the oblique abdominal muscles measured by sEMG in healthy subjects, compared to a sham stimulation.

## Figures and Tables

**Figure 1 jcm-11-03866-f001:**
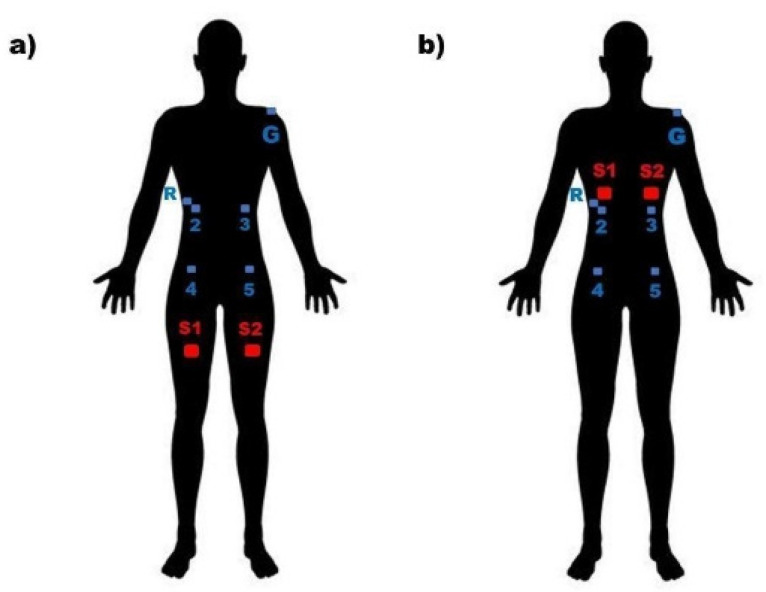
Electrode position. (**a**) Experimental condition non-STI group. (**b**) Experimental condition VSTI group. R: reference electrode; 2: right external oblique electrode; 3: left external oblique electrode; 4: right internal oblique electrode; 5: left internal oblique electrode; G: ground electrode; S1: Stimuli 1 location; S2: Stimuli 2 location.

**Figure 2 jcm-11-03866-f002:**
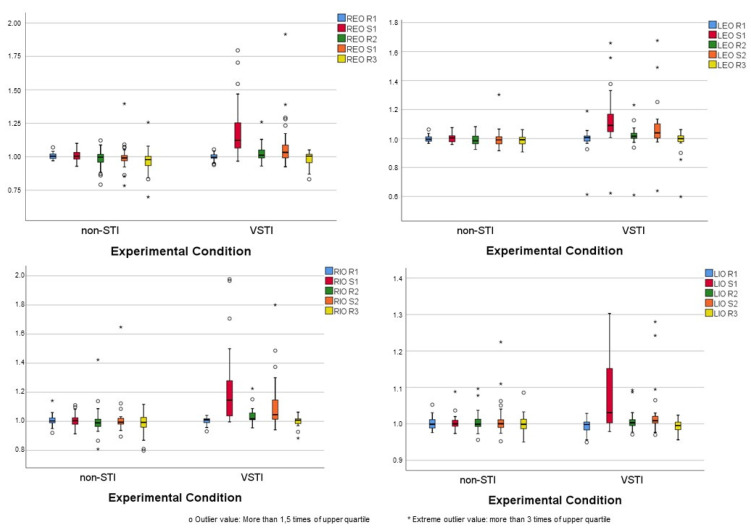
Box plot of electric muscle activity recorded in both experimental conditions in each muscle. REO: Right external oblique; LEO: left external oblique; RIO: right internal oblique; LIO: left internal oblique; R1: repose pre-stimulus; S1: right-side stimulus; R2: rest between S1 and S2; S2: left-side stimulus; R3: repose post-stimulus; non-STI: control experimental condition; VSTI: Vojta experimental condition.

**Table 1 jcm-11-03866-t001:** Descriptive statistics in sample.

	Total Sample *n* = 27
Age (years) *	20.58 (0.72)
Gender (female) **	23 (85.18)
Height (cm) *	164.99 (6.18)
Weight (kg) *	59.92 (9.38)
Body mass index (kg/m^2^) *	21.94 (2.55)

* Mean (standard deviation). ** Count (percentage).

**Table 2 jcm-11-03866-t002:** Electric muscle activity recorded in both experimental conditions.

		Non-STI Condition *n* = 27	VSTI Condition n = 27	*p* *	Hedges’ g
REO	R1	1.003 (0.036)	0.996 (0.026)	0.562	
	S1	1.004 (0.054)	1.123 (0.212)	0.000 **	1.18
	R2	0.997 (0.062)	1.011 (0.065)	0.050	
	S2	0.991 (0.041)	1.033 (0.097)	0.006 **	0.62
	R3	0.978 (0.076)	1.004 (0.067)	0.113	
LEO	R1	0.995 (0.033)	1.007 (0.039)	0.462	
	S1	1.004 (0.045)	1.090 (0.144)	0.000 **	1.01
	R2	0.984 (0.056)	1.015 (0.040)	0.013 *	
	S2	0.991 (0.050)	1.039 (0.102)	0.001 **	0.51
	R3	0.0992 (0.050)	0.999 (0.040)	0.159	
RIO	R1	1.003 (0.044)	1.011 (0.027)	0.634	
	S1	1.005 (0.049)	1.147 (0.247)	0.000 **	1.17
	R2	0.990 (0.052)	1.018 (0.051)	0.001 **	
	S2	0.995 (0.047)	1.047 (0.181)	0.001 **	0.61
	R3	0.993 (0.077)	1.008 (0.035)	0.192	
LIO	R1	0.999 (0.077)	0.998 (0.024)	0.337	
	S1	1.000 (0.018)	1.031 (0.157)	0.000 **	1.06
	R2	0.999 (0.020)	1.003 (0.019)	0.473	
	S2	1.000 (0.022)	1.009 (0.022)	0.065	0.32
	R3	0.999 (0.027)	0.995 (0.024)	0.373	

Medians and interquartile ranges. * Nonparametric tests for independent sample. ** *p*-value < 0.01 Statistically significant. REO: right external oblique; LEO: left external oblique; RIO: right internal oblique; LIO: left internal oblique; R1: repose pre-stimulus; S1: right-side stimulus; R2: rest between S1 and S2; S2: left-side stimulus; R3: repose post-stimulus; non-STI condition: control experimental condition; VSTI condition: Vojta experimental condition.

**Table 3 jcm-11-03866-t003:** Percentage differences found between the phases of each muscle experimental condition.

		Non-STI Condition n = 27		VSTI Condition n = 27		P++ Inter-condition
		% Difference	*p* *	% Difference	*p* *	
REO						
	S1–R1	0.0 (2.9)	0.361	11.7 (20.6)	0.000	0.000
	S2–R2	0.0 (2.5)	0.923	18.3 (9.2)	0.037	0.174
	R3–R1	−3.5 (10.8)	0.008	−0.2 (7.7)	0.239	0.130
	S2–S1	−1.9 (5.4)	0.034	−9.5 (16.3)	0.002	0.030
LEO						
	S1–R1	0.2 (2.5)	0.414	7.2 (16.0)	0.000	0.000
	S2–R2	−0.3 (1.7)	0.701	0.9 (7.7)	0.032	0.110
	R3–R1	−1.5 (2.6)	0.021	0.1 (3.0)	0.361	0.186
	S2–S1	−0.7 (2.0)	0.022	−5.3 (8.5)	0.002	0.033
RIO						
	S1–R1	0.1 (3.3)	0.792	13.1 (23.4)	0.000	0.000
	S2–R2	0.4 (3.1)	0.755	1.2 (16.4)	0.025	0.130
	R3–R1	−1.0 (6.1)	0.118	−0.1 (2.9)	0.829	0.216
	S2–S1	−0.1 (2.8)	0.212	−4.8 (20.0)	0.001	0.008
LIO						
	S1–R1	0.1 (1.2)	0.719	2.6 (15.6)	0.000	0.000
	S2–R2	−0.0 (1.5)	0.866	1.3 (1.7)	0.107	0.149
	R3–R1	0.2 (2.5)	0.981	−0.00 (1.9)	0.904	0.710
	S2–S1	−0.2 (1.2)	0.866	−1.9 (10.7)	0.005	0.012

Medians and interquartile ranges. * Nonparametric tests for independent sample. REO: right external oblique; LEO: left external oblique; RIO: right internal oblique; LIO: left internal oblique; R1: repose pre-stimulus; S1: right-side stimulus; R2: rest between S1 and S2; S2: left-side stimulus; R3: repose post-stimulus; non-STI condition: control experimental condition; VSTI condition: Vojta experimental condition.

## Data Availability

Data are held securely by the research team and may be available upon reasonable request and with relevant approvals in place.

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
