# Peer review of "Electromyographic Response of the Abdominal Muscles and Stabilizers of the Trunk to Reflex Locomotion Therapy (RLT). A Preliminary Study"

_jcm, 2022, doi:10.3390/jcm11133866_

Round 1

Reviewer 1 Report

Daer Autors, congratulations on a good idea for a research paper on the Vojta Method. When preparing for publication, the following considerations should be taken into account:

1. Please, in the sentence "Reflex Crawling Complex = Reflex Rolling is represented in the ontogeny development." Remove the = sign and note that Reflex Crawling is represented in the filogeny development.

2. Please explain what the abbreviations VSTI and STI stand for.

3. Please shorten the number of presented results, and thus decide which results are the most important for a clear explanation of the intended purpose of the work.

4. When presenting applications in an unclear manner, the number 5 appears twice.

5. Please consider the meaning of the sentence "A specific sensory and proprioceptive stimulation on the pectoral area located in the intercostal space in the mammillary line between the seventh and eighth rib, activates the innate muscle responses in the oblique abdominal muscles evaluated by sEMG in healthy subjects, compared to a simulated stimulation according to Vojta Therapy. "

6. Please consider placing the following citation in the paper: Milan Martínek, David Pánek, Tereza Nováková and Dagmar Pavlu. Analysis of Intracerebral Activity during Reflex Locomotion Stimulation According to Vojta's Principle. Applied Sciences 12 (4): 2225 and Wojciech Kiebzak, Arkadiusz Żurawski, Stanisław Głuszek, Michał Koszołowicz, Wioletta Adamus Białek. Cortisol Levels in Infants with Central Coordination Disorders During Vojta Therapy. Children. Children 2021, 8, 1113.

Good luck!

Author Response

Dear Autors, congratulations on a good idea for a research paper on the Vojta Method. When preparing for publication, the following considerations should be taken into account:

  1. Please, in the sentence "Reflex Crawling Complex = Reflex Rolling is represented in the ontogeny development." Remove the = sign and note that Reflex Crawling is represented in the filogeny development.

Thank you very much for your suggestion. We have changed as per you request.

  1. Please explain what the abbreviations VSTI and STI stand for.

Thank you very much for your suggestion. We have explained them.

  1. Please shorten the number of presented results, and thus decide which results are the most important for a clear explanation of the intended purpose of the work.

Thank you for your appreciation. We have removed Cohen’s D and changed Figure 2 into a box plot. We hope it will be better for you.

  1. When presenting applications in an unclear manner, the number 5 appears twice.

Thank you very much. We have re-written the paragraph.

  1. Please consider the meaning of the sentence "A specific sensory and proprioceptive stimulation on the pectoral area located in the intercostal space in the mammillary line between the seventh and eighth rib, activates the innate muscle responses in the oblique abdominal muscles evaluated by sEMG in healthy subjects, compared to a simulated stimulation according to Vojta Therapy. "

Thank you very much. We have changed the meaning of the sentence.

  1. Please consider placing the following citation in the paper: Milan Martínek, David Pánek, Tereza Nováková and Dagmar Pavlu. Analysis of Intracerebral Activity during Reflex Locomotion Stimulation According to Vojta's Principle. Applied Sciences 12 (4): 2225 and Wojciech Kiebzak, Arkadiusz Żurawski, Stanisław Głuszek, Michał Koszołowicz, Wioletta Adamus Białek. Cortisol Levels in Infants with Central Coordination Disorders During Vojta Therapy. Children 2021, 8, 1113.

Thank you very much for your suggestions. Both references have been incorporated into the manuscript.

Good luck!

Reviewer 2 Report

Definitely an interesting topic.

"Electromyographic response of the abdominal muscles and stabilizers of the trunk" - why abdominal muscles and stabilizers of the trunk when only the abdominal obliques were measured?

Terminological ambiguities:

line 43 - postural activity (not reactivity)

line 48 - "decubitus"?

line 63-65, 300-301 - unintelligible

line 200 - Error bars diagrams

Failure to comply with terminology according to Vojta:
reflex rolling/turning, creeping/reptation, pectoral area/point, tactile/digital pressure

femoral rectum and femoral an deltoid rectum muscles?

Lack of explanation of abbreviations:

ECG, MESS, rEMG, SOs

Methodological weaknesses:

"A digital pressure of 1.4 and 1.8 kg /cm2 was exerted ± 200 g [7,8]" - it is not described in the methodology or in the referenced literature sources

"2.4. rEMG Stimuli rEMG Stimuli. "

- EMG type is not described - rEMG and sEMG alternate in the text

- SOs - the abbreviation is not explained and is not used further in the text

- Placement of electrodes at the same distance from the SIAS or umbilicus regardless of the size of the proband? where is the basis for such placement of electrodes?

- Why were only 2 minutes of stimulation measured? What was the basis for this decision?
- Given the non-STI with which all probands started, could there have been no adaptation to position or, conversely, increased sensitivity to the tactile stimulus?

- The results were calculated from the mean change in muscle activity of all probands or from the mean of the differences in muscle activity of all probands?

The discussion lacks justification for not using MVC, which would have greatly simplified the evaluation of the results.

Reference list check to be carried out (mainly Quote 6) and there is a lack of currently published papers on the topic of verifying the effect of Vojta therapy

Author Response

Definitely an interesting topic.

"Electromyographic response of the abdominal muscles and stabilizers of the trunk" - why abdominal muscles and stabilizers of the trunk when only the abdominal obliques were measured?

Thank you for your comment. The oblique muscles, the transversus and muscles found in the back are the main stabilizers of the trunk. Due to the difficulty in recording the transversus through surface electromyography, this muscle was not evaluated. In addition, because of the participant’s position the recording of the back muscles wasn’t possible, and interference was generated because the electrodes were attached to the stretcher. Therefore, the research team finally decided that obliques were the muscles that should be evaluated.

Terminological ambiguities:

line 43 - postural activity (not reactivity)

line 48 - "decubitus"?

line 63-65, 300-301 - unintelligible

line 200 - Error bars diagrams

Thank you very much. We have changed all the terms.

Failure to comply with terminology according to Vojta:
reflex rolling/turning, creeping/reptation, pectoral area/point, tactile/digital pressure femoral rectum and femoral an deltoid rectum muscles?

Thank you very much. We have changed all the terms.

Lack of explanation of abbreviations:

ECG, MESS, rEMG, SOs

Thank you very much. We have added all the explanations.

Methodological weaknesses:

"A digital pressure of 1.4 and 1.8 kg /cm2 was exerted ± 200 g [7,8]" - it is not described in the methodology or in the referenced literature sources

Thank you very much for your comment. We have changed the sentence. We want to indicate that the pressure and direction exerted are the same as reported in the studies of Sanz et al.

"2.4. rEMG Stimuli rEMG Stimuli. "

- EMG type is not described - rEMG and sEMG alternate in the text

Thank you very much. We have clarified the terms in the manuscript.

- SOs - the abbreviation is not explained and is not used further in the text

Thank you very much for your comment. It was a mistake that has been corrected.

- Placement of electrodes at the same distance from the SIAS or umbilicus regardless of the size of the proband? where is the basis for such placement of electrodes?

Thank you very much for your comment. At the beginning, subjects were asked for a muscle voluntary contraction against resistance. The electrodes were placed in the middle third of the total muscle length. In some cases, even bone flange were used. All of them have been detailed in the manuscript.

- Why were only 2 minutes of stimulation measured? What was the basis for this decision?

Thank you for your comment. According to the fundamentals of Vojta Therapy, pressure is exerted between 2 and 7 minutes to achieve muscles and posture effects. The literature shows studies where the pressure was exerted during 2 minutes with positive results on the body. For this reason, 2 minutes were applied in the protocol.

- Given the non-STI with which all probands started, could there have been no adaptation to position or, conversely, increased sensitivity to the tactile stimulus?

Thank you for your comment. The starting position was the same for all subjects. In addition, an adequate time was established to avoid tactile stimuli adaptation, for this reason, both experimental conditions were considered different, and no one affected the other.

- The results were calculated from the mean change in muscle activity of all probands or from the mean of the differences in muscle activity of all probands?

Thank you for your comment. Values were calculated as medians. In Table 2, changes were expressed as the median of one factor from the calculated reference or baseline value in each subject. Table 3 presents the percentage of the difference between each of the recording phases, calculated with the median of the difference in each of the subjects.

The discussion lacks justification for not using MVC, which would have greatly simplified the evaluation of the results.

Thank you very much for your comment. We did not use the maximum voluntary contraction because the aim of our study was to compare the involuntary muscle activity triggered through activation with the Vojta Method vs the basal involuntary muscle activation of the subjects. However, we appreciate your comment and will be take into account for future studies, we are sure that the maximum voluntary muscle activation of each subject will help us to simplify the results.

Reference list check to be carried out (mainly Quote 6) and there is a lack of currently published papers on the topic of verifying the effect of Vojta therapy

Thank you very much. We have added references